# A Weighted and Distributed Algorithm for Range-Based Multi-Hop Localization Using a Newton Method

**DOI:** 10.3390/s21072324

**Published:** 2021-03-26

**Authors:** Jose Diaz-Roman, Boris Mederos, Ernesto Sifuentes, Rafael Gonzalez-Landaeta, Juan Cota-Ruiz

**Affiliations:** 1Department of Electrical and Computer Engineering, Universidad Autónoma de Ciudad Juárez (UACJ), Ciudad Juárez 32310, Mexico; david.roman@uacj.mx (J.D.-R.); esifuent@uacj.mx (E.S.); rafael.gonzalez@uacj.mx (R.G.-L.); 2Department of Physics and Mathematics, Universidad Autónoma de Ciudad Juárez (UACJ), Ciudad Juárez 32310, Mexico; boris.mederos@uacj.mx

**Keywords:** wireless sensor networks, multi-hop weighted localization, robustness

## Abstract

Wireless sensor networks are used in many location-dependent applications. The location of sensor nodes is commonly carried out in a distributed way for energy saving and network robustness, where the handling of these characteristics is still a great challenge. It is very desirable that distributed algorithms invest as few iterations as possible with the highest accuracy on position estimates. This research proposes a range-based and robust localization method, derived from the Newton scheme, that can be applied over isotropic and anisotropic networks in presence of outliers in the pair-wise distance measurements. The algorithm minimizes the error of position estimates using a hop-weighted function and a scaling factor that allows a significant improvement on position estimates in only few iterations. Simulations demonstrate that our proposed algorithm outperforms other similar algorithms under anisotropic networks.

## 1. Introduction

Nowadays, Wireless Sensor Networks (WSNs) have gained relevance in many aspects of our lives because they have the capacity to innovate our environment including body sensor networks, smart houses, automation, transportation, business, security, agriculture to military applications, to mention a few. A WSN is typically composed of a large number of tiny devices called sensor nodes. Due to the characteristics of a sensor node, which contains constrained resources in processing, memory, sensing, and non-renewable with low-power batteries, wireless transmissions must be used efficiently to save energy. Furthermore, even though RF transmissions are around 10 times more energy-expensive than processing, all aspects of energy saving must be carefully taken into account (e.g., leaking energy) to preserve the autonomy of the entire network [1,2,3,4].

Commonly, WSNs are randomly deployed in large geographical areas with the aim of detecting environmental events on sensor nodes. Such events must be reported to other nodes using short-range wireless transmissions until reaching one or more sink nodes for reporting data. Clearly, routing protocols play an important function in multi-hop networks, that is to say, the shortest route from one sensor node to another sensor node allows the entire network to save energy [5,6]. However, it is imperative for data reporting to know where the detected event comes from. In other words, for a broadcast event the time-location parameter is more important than the sensor identity itself. Thus, positions of sensor nodes should be known a priori before to run a WSN application, and so the background stage in most WSN applications is the localization process. In this stage, the geographic-location must be discovered using RF transmissions with neighboring sensors and specific algorithms [7,8,9,10,11]. An interesting research dealing with multi-hop localization is presented in [12]. It manages an hybrid approach where the DV-Hop scheme is improved by promoting unknown sensor as anchor nodes, where these virtual anchors are one-hop neighboring sensor of the real anchors, and their localization are obtained using the estimated distances from the RSSI technique. The authors claim that this scheme can be employed when the density of anchors in the network is low.

In distributed localization algorithms dealing with realistic multi-hop WSN scenarios, estimating among neighboring sensors can suffer from hardware imperfections, changing environments, network topologies, and node densities, which consequently can affect the estimation of the sensor positions [13,14,15,16,17]. For instance, the work in [18] addresses the multi-hop localization problem in severe multi-path scenarios deriving from changing environments. It integrates the non-line-of-sight (NLOS) path information to maximize a reduced-complexity pseudo-maximum likelihood scheme to estimate the position of unknown sensors.

Another interesting approach is proposed in [19] addressing hardware imperfections called wireless ad hoc system for position (WASP). It handles complex radio propagation signals coming from beacons to determine localization of unknown sensors by using novel extensions for ToA measurements. This approach provides high accuracy at minimum hardware and processing cost. It uses a robust least squares algorithm (RLS) for localization that removes iteratively suspicious bad range measurements by recomputing node positions. However, in our work, we pursuit a different approach where the outlierness level of each measurement is controlled by weights that could attenuate the possible influence of atypical observations on the solution of the optimization problem. In [20], the localization problem is addressed using a robust statistic, which reduces the influence of outliers in the estimation process.

The last reasons imply that the accuracy and rate of convergence of range-based iterative algorithms for multi-hop network localization are highly dependent of topologies; therefore, robust approaches must be applied in order to minimize errors in position estimates. Moreover, it is well known that the mathematical formulation of the localization scheme typically requires solving a nonlinear and non-convex optimization problem [21,22,23,24,25].

In this research, a robust and weighted localization algorithm is proposed based on the Newton method approach (RWNM). The algorithm presents the flexibility that any unknown sensor, in a distributed way, can use a dynamic scaling parameter that weights neighboring sensors with the goal of reducing errors in position estimates. Such a scaling parameter is dependent on distance errors with neighboring sensors and the hop-proximity of them towards anchor nodes. Results demonstrated that our approach has a good performance in multi-hop networks with irregular distributions on sensor nodes. It overcomes similar approaches in accuracy on position estimates and requires fewer iterations than others, which is essential in distributed schemes to save energy.

Summarizing, the main contributions of the work are as follows:A novel localization method is introduced that is robust to outliers. To remove or mitigate the influence of possible atypical measurements, the proposed method uses two kind of weights: one is based on the average hop-proximity to the anchors, and the other is based on the determination of the degree of outlierness degree in the noisy distances.The proposed method is based on the Newton method, which improves the unknown sensors’ positions in a few iterations, even though when rough initial estimates are given.The method demonstrates good performance under both isotropic and anisotropic topologies.

The rest of this paper is structured as follows. Section 2 describes the effect of large errors and outliers that could be present in the position estimation process. Section 3 details the problem formulation for multi-hop network scenarios. Section 4 describes the proposed algorithm. Section 5 determines the computational complexity of the algorithm. Section 6 shows some experimental results using isotropic and anisotropic networks, and finally Section 7 draws the main conclusions.

## 2. Reducing the Impact of Outliers in the Estimation Process

Commonly when data $X=\left\{x_{1},x_{2},\ldots ,x_{n}\right\}$ are analyzed and processed, they are assumed as iid (independent and identically distributed) with normal distribution, where the estimation of a statistical parameter μ can be carried out under the Maximum Likelihood Estimator (MLE) technique as
(1)$$\hat{\mu }=\underset{\mu }{\arg\;\min}f\left(X,\mu \right),$$
where
(2)$$f\left(X,\mu \right)=\overset{n}{\prod_{i=1}}\exp^{-\frac{{\left(x_{i}-\mu \right)}^{2}}{2\sigma^{2}}},$$
represents the joint density function for the data set X, μ denotes the mean, and $\sigma^{2}$ the variance. For the estimation of $\hat{\mu }$, the $-\ln f\left(X,\mu \right)$ is applied, which results in the minimization of the sum of square errors as the data have the same dispersion $\sigma^{2}$:(3)$$\hat{\mu }=\underset{\mu }{\arg\;\min}-\ln f\left(X,\mu \right)=\underset{\mu }{\arg\;\min}\overset{n}{\sum_{i=1}}e_{i}^{2},$$
where $e_{i}=x_{i}-\mu$. However, by using the quadratic loss function in (Equation 3), a noticeable amplification of atypical errors (i.e., outliers) is obtained, leading to a bad estimation of μ. The outliers effect can be exemplified with the next case: Assume that $\left\{x_{a},x_{b}\right\}\subset X$ are atypical measurements; thus, (Equation 3) can be rewritten as
(4)$$\hat{\mu }=\underset{\mu }{\arg\;\min}f\left(X,\mu \right)=\underset{\mu }{\arg\;\min}\left(\overset{n}{\sum_{i\notin \{a,b\}}}e_{i}^{2}+e_{a}^{2}+e_{b}^{2}\right).$$

Then, the higher the error values on $x_{a}$ and $x_{b}$, the larger the bias in the estimation of μ. To mitigate this effect, a technique commonly used is to modify the minus log Gaussian likelihood with variances $c_{i}$ depending on each particular regression point as follows:(5)$$f\left(X,\mu \right)=\overset{n}{\prod_{i=1}}\exp^{-\frac{{\left(x_{i}-\mu \right)}^{2}}{2{c_{i}}^{2}}},$$
which implies the following quadratic loss:(6)$$\hat{\mu }=\underset{\mu }{\arg\;\min}-\ln f\left(X,\mu \right)=\underset{\mu }{\arg\;\min}\left(\overset{n}{\sum_{i\notin \{a,b\}}}\frac{e_{i}^{2}}{2{c_{i}}^{2}}+\frac{e_{a}^{2}}{2{c_{a}}^{2}}+\frac{e_{b}^{2}}{2{c_{b}}^{2}}\right),$$
this means that taking $c_{a}$ y $c_{b}$ large enough, then $\frac{e_{a}^{2}}{2{c_{a}}^{2}}+\frac{e_{b}^{2}}{2{c_{b}}^{2}}<<e_{a}^{2}+e_{b}^{2}$; consequently, the outliers at the point *a* and *b* will have less effect in (Equation 6) than in (Equation 4) in the estimation of μ.

## 3. The Range-Based Localization Problem

Consider a set of unknown sensor nodes $S=\{s_{1},s_{2},\ldots ,s_{N}\}$ randomly distributed over a certain large area with 2-D true positions $u_{1}={\left[x_{1},y_{1}\right]}^{T},\;u_{2}={\left[x_{2},y_{2}\right]}^{T},\ldots ,\;u_{N}={\left[x_{N},y_{N}\right]}^{T}$ and estimated positions $p_{1}={\left[{\tilde{x}}_{1},{\tilde{y}}_{1}\right]}^{T},\;p_{2}={\left[{\tilde{x}}_{2},{\tilde{y}}_{2}\right]}^{T},\ldots ,\;p_{N}={\left[{\tilde{x}}_{N},{\tilde{y}}_{N}\right]}^{T}$, respectively. It is known that there is a set $A=\{a_{1},a_{2},\ldots ,a_{M}\}$ of sensor nodes called anchor nodes that are randomly deployed in the same area with a special-hardware to self-localize (i.e., GPS). The anchor nodes have the known positions $q_{N+1}={\left[x_{N+1},y_{N+1}\right]}^{T},\;q_{N+2}={\left[x_{N+2},y_{N+2}\right]}^{T},\ldots ,\;q_{N+M}={\left[x_{N+M},y_{N+M}\right]}^{T}$. It is considered hardware-homogeneity among all N+M deployed sensor nodes with limited radio range R and M≪N.

Under this scenario, a sensor node could require more than one hop to reach neighboring sensors for communications. Then, the set of one-hop neighboring sensors of an unknown sensor $s_{i}$ can be stated as
(7)$$\Omega_{i}=\left\{j\;:\;∥u_{i}-u_{j}∥<R,\;\;\;i\neq j,\;and\;j=1,\ldots ,N\right\},$$
where $∥\cdot ∥$ denotes the Euclidean norm. This research assumes that all unknown sensors have *n*-hops connectivity to anchor nodes in the network, where the distance estimates, $r_{ik}\;$ (for k=1,…,M), between an unknown sensor $s_{i}$ and any anchor $a_{k}$ can be easily calculated by the DV-Hop scheme [13]. Thus, the set of anchor indexes is denoted as B={N+1,N+2,…,N+M}.

The set of noisy distances between $s_{i}$ and its neighboring sensors is described as
(8)$$r_{ij}=d_{ij}+e_{ij}\;\;\;\;\forall \;j\in \Omega_{i},$$
where $d_{ij}=∥u_{i}-u_{j}∥$, $r_{ij}$ represents the noisy distance between sensors $s_{i}$ and $s_{j}$, $d_{ij}$ is the true distance between them, and the term $e_{ij}$ provides the biasing of the estimated distance introduced by environmental conditions and the measurement technique [26]. Distance estimates among sensor nodes can be carried out by techniques like RSS, TDoA, ToA, AoA, or a combination of them [27,28,29]. The localization problem can be formulated as follows:(9)$$\min_{p_{1},p_{2},\ldots ,p_{N}}\sum_{i=1}^{N}(\sum_{j\in \Omega_{i}}f\left(r_{ij}-∥p_{i}-p_{j}∥\right)+\sum_{k\in B}f\left(r_{ik}-∥p_{i}-q_{k}∥\right)),$$
which stands for an unconstrained and nonlinear equation where the function f(·) plays a fundamental role to find position estimates that minimize (Equation 9).

## 4. A Robust and Distributed Localization Algorithm Based on the Newton Method

Due to the high cost in processing data, energy consumption in wireless transmissions, and security of collected data, Equation (Equation 9) tends to be programmed in a distributed way instead of being solved in a central node [30]. The robust-weighted Newton method (RWNM) algorithm, proposed here, breaks down (Equation 9) into subproblems that can be implemented in unknown sensors of a distributed manner [25]. Thus, each sensor $s_{i}$ can minimize its distance error with one-hop neighboring sensors and n-hops anchors by solving the following optimization problem:(10)$$\min_{p_{i}}\;F\left(p_{i}\right)$$
with $F(p_{i})$ defined as
(11)$$F\left(p_{i}\right):=\sum_{j\in \Omega_{i}}f\left(\left(r_{ij}-∥p_{i}-p_{j}∥\right)w_{j}\right)+\sum_{k\in B}f\left(\left(r_{ik}-∥p_{i}-q_{k}∥\right)w_{k}\right).$$

Here, each distance error sensor–sensor is multiplied by a weighted value $w_{j}$, which represents the inverse of the average of the minimum number of hops between the sensor $s_{j}$ and each anchor, stated as
(12)$$w_{j}=\frac{1}{\frac{1}{\left|B\right|}\sum_{k\in B}hops\left(j,k\right)},$$
where hops(j,k) represents the minimum number of hops between the sensor $s_{i}$ and the anchor $a_{k}$. It is easy to define the set of weights corresponding to the neighbors of the sensor $s_{i}$ as
(13)$$W_{i}=\left\{w_{j}:j\in \Omega_{i}\right\}.$$

On the other hand, for sensor–anchor distance errors, the weighted value $w_{k}$ is set to one.

To minimize (Equation 11) in a distributed way, a sensor $s_{i}$ employs the Newton method using the $\ell^{2}$ norm as
(14)$$f\left(e,c\right)=\frac{1}{2}{\left(\frac{e}{c}\right)}^{2},$$
where $e=e_{ih}\cdot w_{h}$, ∀$h\;\in \Omega_{i}\cup B$ stands for either the distance error between sensor–sensor or sensor–anchor, *c* is a scaling factor which controls the broadness of the function shape around the origin and mitigates the effect of large errors as explained in Section 2.

On the other hand, any sensor $s_{i}$ that minimizes position estimates with neighboring sensors must first calculate the scaling parameter $c_{i}$ as described next.

First, using distance errors with neighboring sensors, the median is calculated as
(15)$${\tilde{e}}_{i}=\mathrm{med}\left(E_{\Omega_{i}}\right),$$
where the set of errors $E_{\Omega_{i}}$ is defined as
(16)$$E_{\Omega_{i}}=\left\{\right|r_{ij}-∥p_{i}-p_{j}∥|:j\in \Omega_{i}\},$$
then, a new set of neighboring sensor of $s_{i}$ that presents less error in distance estimates than ${\tilde{e}}_{i}$ is obtained as
(17)$${\tilde{\Omega }}_{i}=\left\{j\in \Omega_{i}:\left|r_{ij}-∥p_{i}-p_{j}∥\right|\leq {\tilde{e}}_{i}\right\}.$$

Next, using (Equation 13) with the new set of neighbors of (Equation 17)$,{\tilde{W}}_{i}=\left\{w_{j}:j\in {\tilde{\Omega }}_{i}\right\}$, the maximum weight value is obtained in Equation (Equation 18):(18)$$w_{max_{i}}=\max_{w\in \tilde{W_{i}}}w.$$

Finally, the scaling parameter vector for neighboring sensors of $s_{i}$ and anchor nodes is computed according to Equation (Equation 19):(19)$$c_{j}=\left\{\begin{array}{cc} w_{max_{i}}, & j\;\in \;{\tilde{\Omega }}_{i},\\ \;\;\infty , & j\;\in \;\Omega_{i}\backslash {\tilde{\Omega }}_{i},\\ 1, & j\;\in \;B. \end{array}\right.$$

It is remarkable to mention that $c_{j}=\infty$, which implies that the sensor $s_{j}$ will be discarded in the minimization process.

The Hessian and the gradient for the Newton method process are described as follows. For clarity, consider that the position $p_{r}$ can be used indistinctly for both anchor and unknown sensor positions. The gradient of (Equation 11), $\nabla F\left(p_{i}\right)={\left(\frac{\partial F(p_{i})}{\partial x_{i}},\frac{\partial F(p_{i})}{\partial y_{i}}\right)}^{T}$ is stated as
(20)$$\frac{\partial F(p_{i})}{\partial x_{i}}=-\sum_{r\in \Omega_{i}\cup B}\left(\frac{\partial f}{\partial e_{ir}}\right)\frac{\left(x_{i}-x_{r}\right)}{∥p_{i}-p_{r}∥},$$
(21)$$\frac{\partial F(p_{i})}{\partial y_{i}}=-\sum_{r\in \Omega_{i}\cup B}\left(\frac{\partial f}{\partial e_{ir}}\right)\frac{\left(y{}_{i}-y{}_{r}\right)}{∥p_{i}-p_{r}∥}$$
where
(22)$$\frac{\partial f}{\partial e_{ir}}=\frac{e_{ir}\cdot w_{r}^{2}}{c_{r}^{2}}.$$

The Hessian of (Equation 11) is described as
(23)$$H\left(p_{i}\right)=\left[\begin{array}{cc} \frac{\partial F(p_{i})}{\partial^{2}x_{i}} & \frac{\partial F(p_{i})}{\partial x_{i}\partial y_{i}}\\ \frac{\partial F(p_{i})}{\partial y_{i}\partial x_{i}} & \frac{\partial F(p_{i})}{\partial^{2}y_{i}} \end{array}\right],$$
(24)$$\begin{array}{c} \frac{\partial F(p_{i})}{\partial^{2}x_{i}}=\sum_{r\in \Omega_{i}\cup B}[\left(\frac{\partial f}{\partial e_{ir}}\right)\left[\begin{array}{cc} \frac{{\left(x_{i}-x_{r}\right)}^{2}}{{\left[{\left(x_{i}-x_{r}\right)}^{2}+{\left(y_{i}-y_{r}\right)}^{2}\right]}^{3\;/\;2}}- & \frac{1}{{\left[{\left(x_{i}-x_{r}\right)}^{2}+{\left(y_{i}-y_{r}\right)}^{2}\right]}^{1\;/\;2}} \end{array}\right]+\\ \left(\frac{\partial^{2}f}{\partial e_{ir}^{2}}\right)\left[\frac{{\left(x_{i}-x_{r}\right)}^{2}}{{\left(x_{i}-x_{r}\right)}^{2}+{\left(y_{i}-y_{r}\right)}^{2}}\right]], \end{array}$$
(25)$$\begin{array}{c} \frac{\partial F(p_{i})}{\partial^{2}y_{i}}=\sum_{r\in \Omega_{i}\cup B}[\left(\frac{\partial f}{\partial e_{ir}}\right)\left[\begin{array}{cc} \frac{{\left(y{}_{i}-y_{r}\right)}^{2}}{{\left[{\left(x_{i}-x_{r}\right)}^{2}+{\left(y_{i}-y_{r}\right)}^{2}\right]}^{3\;/\;2}}- & \frac{1}{{\left[{\left(x_{i}-x_{r}\right)}^{2}+{\left(y_{i}-y_{r}\right)}^{2}\right]}^{1\;/\;2}} \end{array}\right]+\\ \left(\frac{\partial^{2}f}{\partial e_{ir}^{2}}\right)\left[\frac{{\left(y{}_{i}-y_{r}\right)}^{2}}{{\left(x_{i}-x_{r}\right)}^{2}+{\left(y_{i}-y_{r}\right)}^{2}}\right]], \end{array}$$
(26)$$\frac{\partial F(p_{i})}{\partial x_{i}\partial y_{i}}=\sum_{r\in \Omega_{i}\cup B}\left[\left(\frac{\partial f}{\partial e_{ir}}\right)\left[\begin{array}{c} \frac{\left(x_{i}-x_{r}\right)\left(y_{i}-y_{r}\right)}{{\left[{\left(x_{i}-x_{r}\right)}^{2}+{\left(y_{i}-y_{r}\right)}^{2}\right]}^{3\;/\;2}} \end{array}\right]+\left(\frac{\partial^{2}f}{\partial e_{ir}^{2}}\right)\left[\frac{\left(x_{i}-x_{r}\right)\left(y_{i}-y_{r}\right)}{{\left(x_{i}-x_{r}\right)}^{2}+{\left(y_{i}-y_{r}\right)}^{2}}\right]\right],$$
(27)$$\frac{\partial F(p_{i})}{\partial y_{i}\partial x_{i}}=\sum_{r\in \Omega_{i}\cup B}\left[\left(\frac{\partial f}{\partial e_{ir}}\right)\left[\begin{array}{c} \frac{\left(y_{i}-y_{r}\right)\left(x_{i}-x_{r}\right)}{{\left[{\left(x_{i}-x_{r}\right)}^{2}+{\left(y_{i}-y_{r}\right)}^{2}\right]}^{3\;/\;2}} \end{array}\right]+\left(\frac{\partial^{2}f}{\partial e_{ir}^{2}}\right)\left[\frac{\left(y_{i}-y_{r}\right)\left(x_{i}-x_{r}\right)}{{\left(x_{i}-x_{r}\right)}^{2}+{\left(y_{i}-y_{r}\right)}^{2}}\right]\right],$$
where
(28)$$\frac{\partial^{2}f}{\partial e_{ir}^{2}}=\frac{w_{r}^{2}}{c_{r}^{2}}.$$

Assuming that rough initial estimates for all unknown sensor nodes are obtained by standard methods such as Least-Squares and Min-Max, among others [11], the distributed localization process for a sensor $s_{i}$ follows the process shown in Algorithm 1.
**Algorithm 1** Sensor $s_{i}$ refining its position estimates.**Input**: $p_{i},W_{i},\mathrm{Niter},\Omega_{i},B$**Output**: Refined version of $p_{i}$ 1: **Initialize**: $\rho =0.05,\tau =10^{-2}$ 2: **Compute**: $c_{r}$ for all $r\in \Omega_{i}\cup B$ from (Equation 19) 3: *n* = 0 4: **repeat** 5:  ${\tilde{p}}_{i}=p_{i}$ 6:  Compute: $\nabla F({\tilde{p}}_{i})$ from (Equation 20) and (Equation 21) 7:  Compute: $H({\tilde{p}}_{i})$ from (Equation 23) 8:  $\mu =\|\nabla F({\tilde{p}}_{i})\|\rho$ 9:  Solve: $\left(H({\tilde{p}}_{i})+\mu I)\Delta =-\nabla F({\tilde{p}}_{i}\right)$ 10:   $p_{i}={\tilde{p}}_{i}+\alpha \Delta$ 11:  n=n+1 12: **until**
$\left(n\leq \mathrm{Niter}\mathrm{or}\|p_{i}-{\tilde{p}}_{i}\|\leq \tau \right)$

As is well known, the optimal step length α in line 10 of Algorithm 1 can be found using both the Armijo and Wolfe conditions at expenses of more computational cost [31,32]. For practical purposes, α was set to one. The updated position $p_{i}$ of $s_{i}$ is broadcast to its neighboring sensors starting a new refining position process.

## 5. Computational Complexity

The Algorithm 1 is run for each sensor $s_{i}$ and requires as input the set $W_{i}=\left\{w_{j}\;:\;j\in \Omega_{i}\right\}$ and the set of weights $C_{i}=\left\{c_{j}\;:\;j\in \Omega_{i}\cup B\right\}$ computed in Equation (Equation 19). First, the computational cost incurred in the computation of the $W_{i}$ and $C_{i}$ will be determined.

As each $w_{j}\in W_{i}$ is computed as the average of the number of hops hops(j,k) between the sensor $s_{j}$ and each anchor *k*, then computing $w_{j}$ takes O(|B|) operations. On the other hand, the set $W_{i}$ consists of $|\Omega_{i}|$ elements, therefore the cost to determine the set $W_{i}$ is
(29)$$\cos t\left(W_{i}\right)=O(|\Omega_{i}|\times |B|).$$

Each weight $c_{j}\in C_{i}$ is calculated by Equation (Equation 19) which depends on (Equation 17) corresponding to the median of a set $E_{\Omega_{i}}$ of size at most $|\Omega_{i}|$. It is well known [33] that the median of a set of size *n* can be computed in linear time O(n). Therefore, the computational cost to determine $C_{i}$ is
(30)$$\cos t\left(C_{i}\right)=O(|\Omega_{i}\left|+|B|\right)=O(|\Omega_{i}\left|\right),$$
assuming that $\left|B|\leq \right|\Omega_{i}|$.

In the following, the computational cost of one iteration of the repeat/until block (from line 5 to 11) will be addressed. The most expensive steps are the calculus of the gradient $\nabla F({\tilde{p}}_{i})$ and the Hessian $H({\tilde{p}}_{i})$, where each of them are computed in at most $O(|\Omega_{i}\left|+|B|\right)=O(|\Omega_{i}\left|\right)$. This is due to both the first and second partial derivatives (equations from (Equation 20) to (Equation 27)) are expressed as summation over the set $\Omega_{i}\cup B$. Note that line 9 corresponds to solving a 2×2 linear system with a positive definite matrix with cost O(1). Summarizing, the cost of this loop block is
(31)$$\cos t(\mathrm{repeat}/\mathrm{until})=O(n_{i}|\Omega_{i}|),$$
where $n_{i}$ is the number of iterations that the algorithm needed to convergence. Therefore, the total cost of the Algorithm 1 per sensor $s_{i}$ is calculate as
(32)$$\mathrm{cost}\left(W_{i}\right)+\mathrm{cost}\left(C_{i}\right)+\mathrm{cost}(\mathrm{repeat}/\mathrm{until})=O(|\Omega_{i}|\times |B|)+O(n_{i}|\Omega_{i}\left|\right).$$

Because Algorithm 1 is executed once for each sensor, it is obtained that
(33)$$\mathrm{final}\;\mathrm{cost}=\sum_{i\;:\;s_{i}\in S}\left[O(\right|\Omega_{i}|\times |B|)+O(n_{i}|\Omega_{i}|)].$$

Based on the experimental analysis, |B|=5 and the body of the repeat until block is executed in 6 iterations ($n_{i}=6$) on average. Consequently, Equation (Equation 33) can be simplified to
(34)$$\mathrm{final}\;\cos t=\sum_{i\;:\;s_{i}\in S}\left[O(5\right|\Omega_{i}\left|\right)+O(6|\Omega_{i}\left|)\right]=\sum_{i\;:\;s_{i}\in S}O(|\Omega_{i}\left|\right).$$

As $\sum_{i\;:\;s_{i}\in S}|\Omega_{i}\left|=2|E\right|$, with |E| denoting the number of wireless connections, the final cost is a big-*O* of the number of connections in the network, implying that the algorithm scales linearly with respect to the total number of connections.

## 6. Range-Based Multi-Hop Network Performance

In large deployment scenarios with constrained radio range on sensor nodes, forming multi-hop networks, and unknown sensors far outnumber anchor nodes, the DV-Hop scheme represents a good approach to estimate distances among sensor nodes in a distributed manner [11,13]. As is well known, a distance estimation between an anchor node and unknown sensor, at some hops of each other, normally could suffer from several error sources: the environmental itself, the ranging technique, and the localization algorithm, to name a few. Furthermore, the network topology, integrated to the other aforementioned factors, can generate abnormal errors (i.e., outliers) on distance estimates; the latter can bring with them large errors on distance estimates where most localization algorithms do not behave well.

To evaluate the accuracy iterations performance of algorithms, two kind of topologies are recreated as shown Figure 1. As can be observed, both networks are connected (i.e., non-isolated sensor nodes), and anchor nodes are distributed around a circle shape Figure 1a (anisotropic network) or inside of a square area Figure 1b (isotropic network). It must be remarked that anchor nodes are located in a non-collinear pattern distribution.

The system test has 10 setup networks for each topology with fixed positions of sensor and anchor nodes, and the degree of network connectivity is given by the radio range in sensor and anchor nodes, assumed homogeneous. There are two levels of degree connectivity given by different radio ranges (R = 35 m and R = 45 m). Therefore, for each radio range, 10 networks are created for each topology.

As proposed in [34], the noisy distances between a sensor $s_{i}$ and its neighbors are computed as
(35)$$r_{ij}=∥u_{i}-u_{j}∥\cdot \max\left\{0,1+\chi \cdot nfe\right\},\;\;\forall j\in \Omega_{i},$$
where χ∼N(0,1) represents a normal distributed random variable and nfe∈[0,1] is the standard deviation of the distance error. This is equivalent to
(36)$$r_{ij}=\max\left\{0,∥u_{i}-u_{j}∥+∥u_{i}-u_{j}∥\chi \cdot nfe\right\},\;\;\forall j\in \Omega_{i},$$
that can be rewritten as
(37)$$r_{ij}=\max\left\{0,d_{ij}+e_{ij}\right\},\;\;\forall j\in \Omega_{i},$$
which is similar to the additive noise model in (Equation 8).

In order to obtain noisy distances contaminated with outliers, a parameter Θ (percentage of outliers) is used to randomly select a subset of distances. Each distance in the subset is modified by a scalar parameter ρ as follows:(38)$$r_{ij}=∥u_{i}-u_{j}∥\cdot \max\left\{0,1+\chi \cdot nfe\right\}\cdot \rho ,\;\;\forall j\in \Omega_{i}.$$
where the parameter ρ is defined as
(39)$$\rho =\left\{\begin{array}{cc} 5, & \mathrm{if}\;\;\;\max[0,1+\chi \cdot nfe]\geq 1,\\ \frac{1}{5}, & \mathrm{if}\;\;\;\max[0,1+\chi \cdot nfe]<1, \end{array}\right.$$

For each sensor $s_{i}$ an initial estimated position is calculated as the average of the *M* anchor positions plus a random perturbation as follows:(40)$$\frac{1}{M}\overset{M}{\sum_{k=1}}\left(x_{N+k}+\upsilon_{i,k},\;y_{N+k}+\upsilon_{i,k}\right),$$
where $\upsilon_{i,k},i=1,\ldots ,N,k=1,\ldots ,M$ is a N×M sample of a random variable with Gaussian distribution N(0,1).

The root mean square error (RMSE) metric is employed to evaluate errors on position estimates of the unknown sensors for the network *t*, where t=1,…,10, and finally the average RMSE (RMSEavg) is calculated as follows:(41)$$RMSEavg=\frac{1}{10}\overset{10}{\sum_{t=1}}\left[\sqrt{\frac{1}{N}\overset{N}{\sum_{i=1}}{∥p_{i}^{t}-u_{i}^{t}∥}^{2}}\;\;\right].$$

In Equation (Equation 41), $∥\cdot ∥$ is the two-norm, $p_{i}^{t}$ and $u_{i}^{t}$ represent the position estimate of sensor $s_{i}$ and its true position for the network *t*, respectively, and *N* is the number of unknown sensors.

Figure 2 shows the methodology used for the algorithms to evaluate accuracy and number of iterations. There are two ranges of connectivity for testing (R = 35 m and R = 45 m), two levels of noise (nfe=0.1 and nfe=0.3), and six scales of outliers (Θ∈{0%,10%,20%,30%,40%,50%}). The box on the left side, in Figure 2, has four possible combinations between R and nfe values. For each one of the four combinations, 30 evaluation processes are generated as shown the box on the right side. For instance, the remarked arrows, left side, imply that the combination R = 35 m with nfe=0.1 will be applied to all distance measurements on an entire network. Using this combination (R = 35 m and nfe=0.1), without outliers (Θ=0%), all iterative algorithms are tested using the 10 independent networks. For each one of the 10 networks, the algorithm is run until one of the stopping criteria is satisfied (i.e., 100 iterations or $∥p_{i}-{\tilde{p}}_{i}∥\leq \tau$). Then, the number of iterations in which the algorithm stopped is registered. Finally, Equation (Equation 41) is used to calculate the average RMSE (RMSEavg) of the 10 networks and also the average of the numbers of iterations (Iter_avg) is obtained. The next step consists of testing the same combination (R = 35 m and nfe=0.1), but now adding 10% of outliers to noisy distances with (Equation 38). This process is repeated until reaching 50% of outliers in noisy distances. Therefore, for each outlier step there will be two parameters (i.e., RMSEavg and Iter_avg) for each evaluated algorithm. The last process is repeated for R = 35 m with nfe=0.3, R = 45 m with nfe=0.1 and finally for R = 45 m with nfe=0.3. Table 1 summarizes the parameters used in the experimental simulations.

### 6.1. Range-Based Multi-Hop Localization over Randomly and Uniformly Distributed Sensor Networks

In this section, we test the proposed RWNM algorithm under isotropic WSN topologies. Here, sensor nodes are uniformly and randomly distributed over an area of 200×200 m${}^{2}$. Ten multi-hop networks similar to that one presented in Figure 1b are created. The ratio between the number of unknown sensors and anchors is 0.95. Each network has 100 sensor nodes. First, it is assumed that each sensor node has a partial set of neighboring sensors according to Equation (Equation 7). Furthermore, anchors and unknown sensors are assumed with the same circular radio range R. Each pairwise distance is affected with a noise factor of nfe=0.1 using Equation (Equation 35). This benchmark network, with nfe=0.1 and Θ=0, is used as a basis to create other five networks, affected with different levels of outliers Θ=10%,Θ=20%,Θ=30%,Θ=40%, and Θ=50%. Finally, each one of the six networks is run in every iterative algorithm to evaluate its accuracy and rate of convergence (or iterations) at different levels of outliers. It must be remarked that DV-hop [13] and RELM [35] algorithms modify distances between anchors and unknown sensors, where such distances are unaffected by noise or by outliers as they depend on hop counts between them. Initial estimates average of the 10 networks is around 86.33 m.

It is known that there is at least one path between any pair of sensor nodes in each network. Initial position estimates for the iterative algorithms are simply calculated as the average positions of anchor nodes. The refinement stage is run in a distributed way as second localization process in each unknown sensor to improve position estimates. Figure 3 shows the accuracy iteration behavior of the evaluated algorithms considering nfe=0.1 and R = 35 m at different levels of outliers.

Clearly, the accuracy performance of the combination algorithms RELM [35] +LM [36] and DV-Hop+LM are linearly affected with the increase of outliers on the range measurements among sensor nodes. However, without presence of outliers, the best accuracy on position estimates is obtained by the DV-Hop+LM algorithm with a RMSEavg = 21.83 m. The worst RMSEavg-Iter_avg performance is carried out by the combination RELM [35] +SOCP [34,37] algorithms. This is because the SOCP algorithm is highly dependent on the number and location of the anchors. On average, the DV-Hop+DWDSCL [38] obtains the best accuracy performance with 23.36 m followed by the combination DV-Hop-RWNM with 24.04 m. Moreover, this last combination outperforms others in rate of convergence with only 4.25 iterations, in average, which implies more saving energy for the entire network as shown Table 2.

Now, keeping both the position set of unknown sensors and anchors positions for each one of the 10 benchmark networks and also taking the same set of range connections (i.e., R = 35 m), the distance errors among neighboring sensors are now affected with nfe=0.3, and the results are depicted in Figure 4.

The results show a RMSEavg-Iter_avg behavior very similar to the previous one. The DV-HOP+RWNM scheme still manages to be lower in iterations with 4.06, in average. The average results are shown in Table 3.

On the other hand, increasing the radio range from R = 35 m to R = 45 m, in each sensor node, produces more gathered information of neighboring sensors, and it also presents a reduction in the number of hops between anchors and unknown sensors. Both last effects tend to reduce error on position estimates. Figure 5 shows results with nfe=0.1 and R = 45 m.

It is clear that algorithms reduce the error on position estimates, on average, as shown in Table 4. Figure 5 demonstrates how errors on position estimates tend to increase when the quantity of outliers also increase, as expected. Without outliers (i.e., Θ=0%) on distance estimates, the DV-Hop+LM has the best accuracy performance with RMSEavg = 9.72 m, but it increases its error as outliers do. On the other hand, due to the weighted function in the their minimization process, the DV-HOP+DWDSCL and DV-HOP+RWNM schemes are less affected in the RMSEavg, and the RWNM approach requires around 5.05 iterations, on average, to obtain the best RMSEavg result of 14.78 m.

When the range distance errors among neighboring sensor are increased from nfe=0.1 to nfe=0.3, errors on position estimates are also slightly increased as shown Figure 6 and Table 5. It must remarked that the combination DV-Hop-RWNM still hold the best performance in accuracy and iterations with 15.22 m and 5.45, respectively.

Figure 7 illustrates the behavior of the RMSEavg versus number of iterations considering two radio ranges. It can be remarked that the proposed scheme shows a good performance through iterations. It reaches an average RMSE of 23.15 m in 3 iterations for R = 35 m and an average RMSE of 14.29 m also in 3 iterations for R = 45 m.

### 6.2. Range-Based Multi-Hop Localization over Irregular Topologies of Sensor Networks

This section analyzes the accuracy and iterations performance of the proposed RWNM scheme under anisotropic networks. A benchmark of 10 independent multi-hop networks like that presented in Figure 1a is considered for testing. In all networks, it is known that there is at least one path between every pair of nodes around the circle shape.

The analysis and tests for this section run also in the same way as that analyzed in Section 6.1. The first test consists of evaluating the error on position estimates and the number of iterations spent by the distributed algorithms. The stopping criterion is assumed equal in all algorithms (i.e., $\tau =10^{-2}$). Initial estimates average of the 10 networks is around 102.1 m. Using the combination nfe and R, 10 independent networks are created, and each one is run in the distributed algorithms to finally averaging the results (i.e., RMSEavg and Iter_avg). Figure 8 shows the results for nfe = 0.1 and R = 35 m.

Despite having a large initial error on position estimates (i.e., ~102.1 m), most algorithms (except the SOCP scheme) end up with a RMSEavg below to 22 m of error on position estimates without presence of outliers (i.e., Θ = 0%). The combinations RELM+LM and DV-Hop+LM are linearly affected with increasing outliers. The combination RELM+SOCP is unaffected by outliers, but its RMSEavg is too high compared with the remaining four combinations. For example, when there are not outliers (i.e., Θ = 0%) in the estimated distances between sensors, the best result of the estimated positions is obtained by the combination DV-Hop+LM reaching a RMSEavg = 15.24 m followed by the DV-Hop+RWNM scheme with 16.25 m. However, the accuracy on position estimates affects the former method linearly with the increase of outliers, providing a RMSEavg = 31.06 m at Θ = 50%. On the other hand, the combination DV-Hop+RWNM is less affected by the presence of outliers. It ends up with a RMSEavg = 16.94 m for a Θ = 50%. It should also be recognized that the best RMSEavg-Iter_avg performance is obtained by this combination, as shown the Table 6 with only 4.15 iterations and a RMSE = 16.49 m, on average.

The next step is to increase the noise level on distance estimates from nfe=0.1 to nfe=0.3 in each one of the 10 benchmark networks and carry out the test like in the previous step. Figure 9 shows that all combinations almost have the same behavior. The RMSEavg of the combination DV-Hop+RWNM slightly increases as the quantity of outliers also does. It starts with a RMSEavg = 16.47 m at Θ = 0% and finishes with RMSEavg = 17.19 m at Θ = 50%. Furthermore, this combination gets the best trade-off between RMSEavg and Iter_avg as summarized in Table 7.

Another interesting test to analyze consists of increasing the radio range in sensor nodes; so the radio range is set to R = 45 m with nfe=0.1 in all sensor nodes in the 10 independent benchmark networks. Figure 10a shows the accuracy of the estimated positions for this case.

As expected, increasing the radio range improves the accuracy of most algorithms at Θ=0%. However, as an outlier could be present in a range measurement, it will affect more neighboring sensors in the estimation process, and most algorithms tend to increase their errors as outliers do. The proposed scheme RWNM starts with a RMSEavg = 16.29 m at Θ=0% and finishes with a RMSEavg = 16.39 m at Θ=50%. Table 8 shows that the combination DV-Hop+RWNM continues with the best results for accuracy and iterations with a RMSEavg = 16.24 m and 5.33 iterations on average.

The final step uses the 10 benchmark networks with R = 45 m and nfe=0.3. Even though the noise level has increased in distance measurements, the final results in both analyzed parameters (i.e., RMSEavg and Iter_avg) remain close similar to the last step as depicted Figure 11. The combinations DV-Hop+RWNM continue with the best accuracy performance with the minimum number of iterations as depicted in Table 9.

Figure 12 depicts the relation between the RMSEavg and number of iterations considering two radio ranges for anisotropic networks. It can be observed that the proposed scheme provides the best accuracy and iterations performance. It can be appreciate that the DV-Hop+RWNM algorithms spends only three iterations to reach a RMSEavg of 17.01 m and 16.2 m for R = 35 m and R = 45 m, respectively.

## 7. Conclusions

This research has investigated a range-based and distributed algorithm suitable for multi-hop network localization. The proposed scheme employs a weighted Newton method approach that minimizes errors in position estimates under anomalous pairwise range measurements between sensor nodes. Unknown sensors categorize neighboring nodes with specific weights based on the closeness to an anchor node and error distances. The algorithm has been tested at different radio ranges and noise levels in distance estimates, demonstrating robustness in presence of outliers with a small number of iterations to reach good results in accuracy. In fact, in isotropic networks, the proposed method practically matches the localization performance of the DV-HOP+DWDSCL method using far fewer iterations to converge. The best performance is carried out under anisotropic networks where the proposed scheme outperforms others providing the best accuracy in position estimates with the minimum number of iterations.

## Figures and Tables

**Figure 1 sensors-21-02324-f001:**
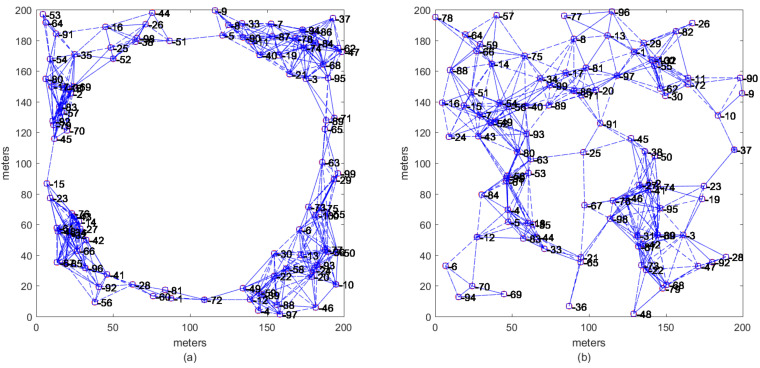
Multi-hop WSN topologies, where each connected network contains 95 unlocalized sensors and five anchor nodes. (**a**) Anisotropic circular connected network with sensor and anchor nodes randomly distributed in a circular shape. (**b**) Isotropic uniform distributed network with sensor and anchor nodes randomly distributed over a square area of 40,000 m${}^{2}$.

**Figure 2 sensors-21-02324-f002:**
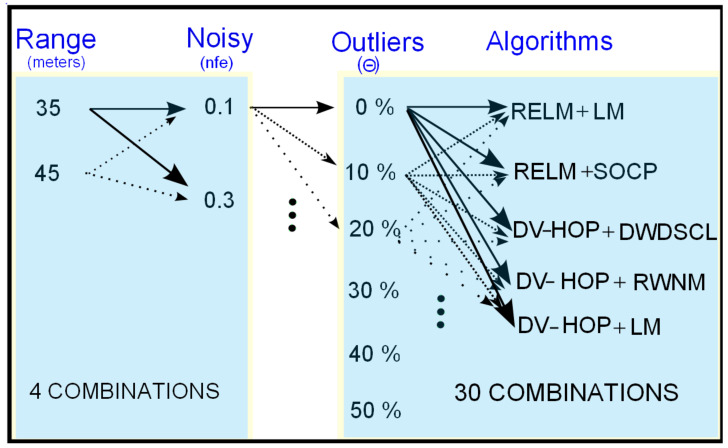
Methodology followed to test RMSEavg and Iter_avg spent by algorithms using a setup of 10 networks for both circular and uniform node distributions.

**Figure 3 sensors-21-02324-f003:**
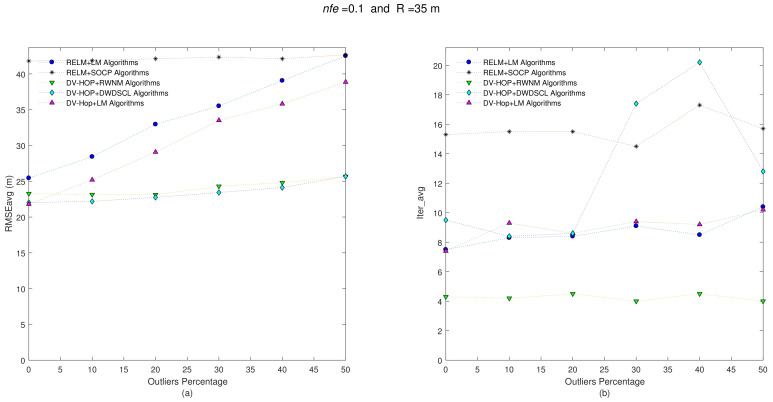
(**a**) RMSEavg and (**b**) Iter_avg after running five iterative algorithms over 10 multi-hop isotropic networks with R = 35 m and nfe=0.1. Outliers levels are increased by steps of 10% from 0% to 50%.

**Figure 4 sensors-21-02324-f004:**
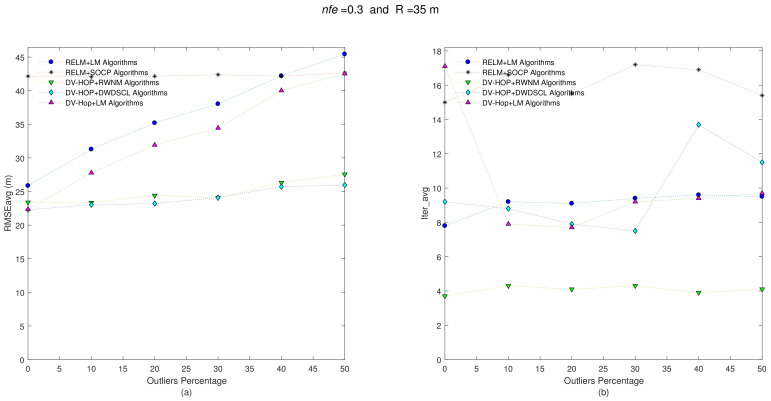
(**a**) RMSEavg and (**b**) Iter_avg after running five iterative algorithms over 10 multi-hop isotropic networks with R = 35 m and nfe=0.3. Outliers levels are increased by steps of 10% from 0% to 50%.

**Figure 5 sensors-21-02324-f005:**
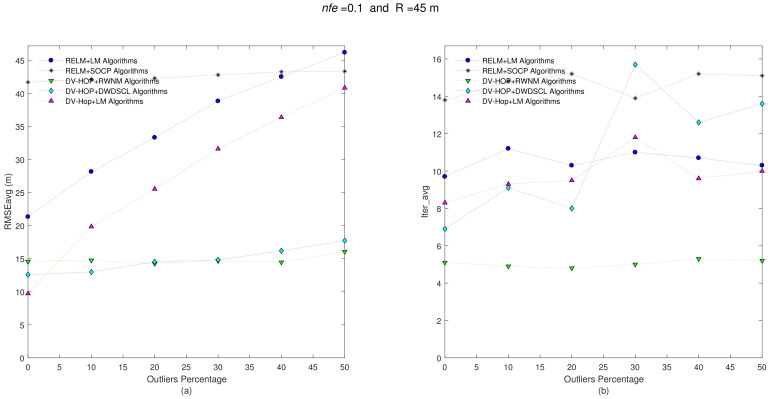
(**a**) RMSEavg and (**b**) Iter_avg after running five iterative algorithms over 10 multi-hop isotropic networks with R=45 m and nfe=0.1. Outliers levels are increased by steps of 10% from 0% to 50%.

**Figure 6 sensors-21-02324-f006:**
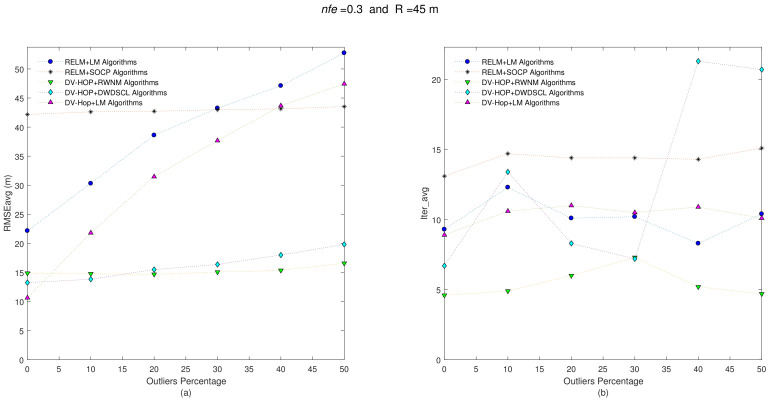
(**a**) RMSEavg and (**b**) Iter_avg after running five iterative algorithms over 10 multi-hop isotropic networks with R=45 m and nfe=0.3. Outliers levels are increased by steps of 10% from 0% to 50%.

**Figure 7 sensors-21-02324-f007:**
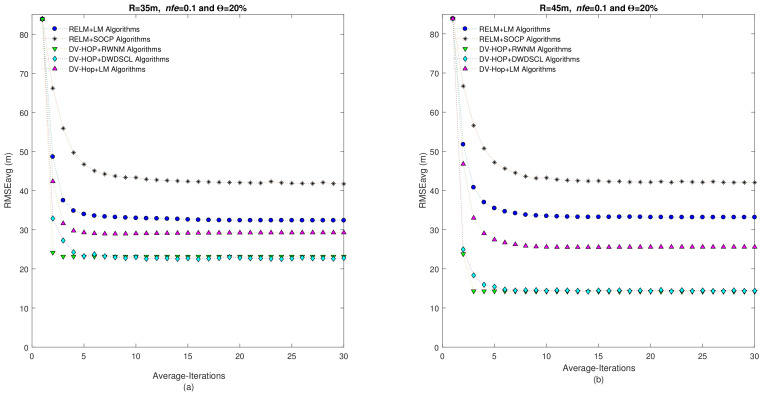
RMSEavg vs. Iterations of algorithms using Θ=20% and nfe=0.1 considering the average of 10 independent isotropic networks. (**a**) Corresponds to R = 35 m and (**b**) R = 45 m.

**Figure 8 sensors-21-02324-f008:**
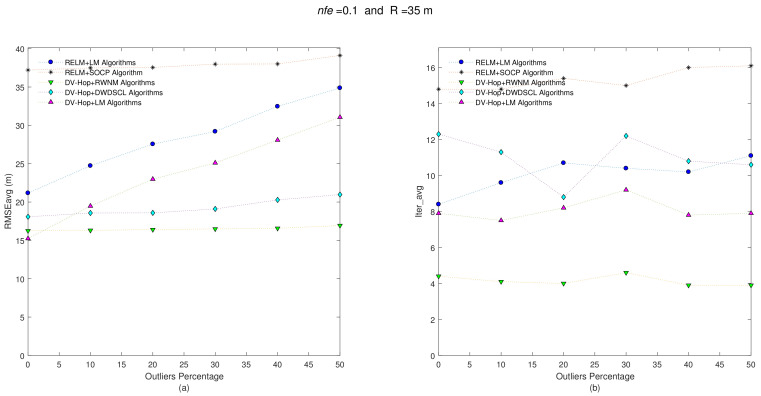
(**a**) RMSEavg and (**b**) Iter_avg after running five iterative algorithms over 10 multi-hop anisotropic networks with R = 35 m and nfe=0.1. Outliers levels are increased by steps of 10% from 0% to 50%.

**Figure 9 sensors-21-02324-f009:**
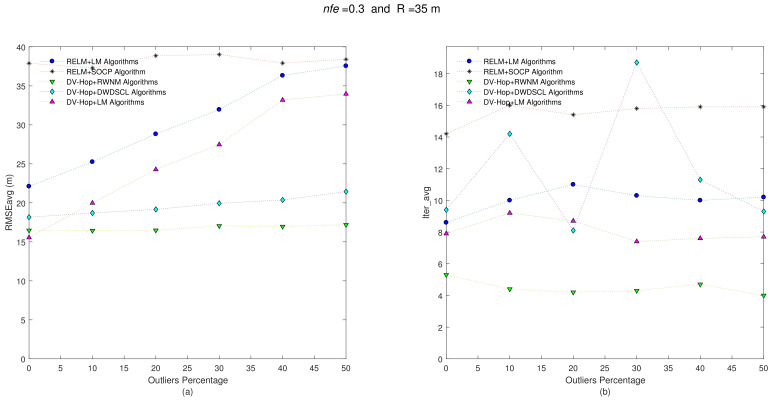
(**a**) RMSEavg and (**b**) Iter_avg after running five iterative algorithms over 10 multi-hop anisotropic networks with R = 35 m and nfe=0.3. Outliers levels are increased by steps of 10% from 0% to 50%.

**Figure 10 sensors-21-02324-f010:**
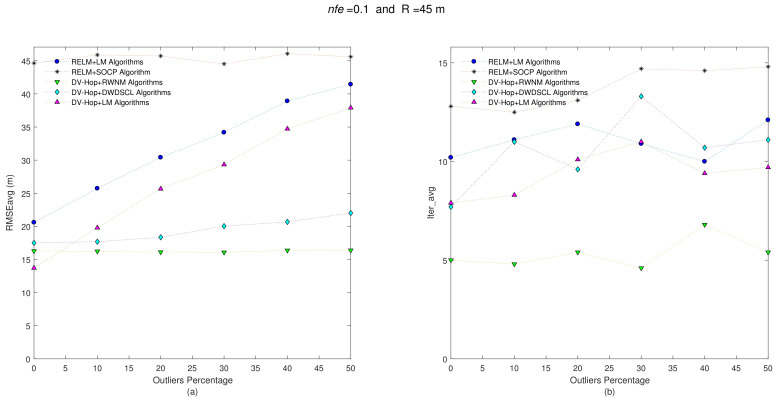
(**a**) RMSEavg and (**b**) Iter_avg after running five iterative algorithms over 10 multi-hop anisotropic networks with R = 45 m and nfe=0.1. Outliers levels are increased by steps of 10% from 0% to 50%.

**Figure 11 sensors-21-02324-f011:**
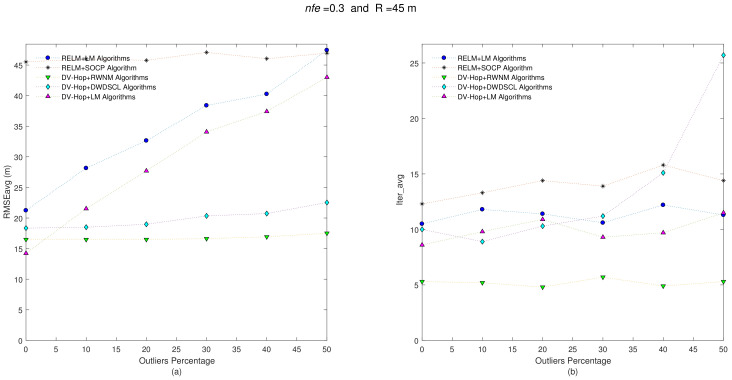
(**a**) RMSEavg and (**b**) Iter_avg after running five iterative algorithms over 10 multi-hop anisotropic networks with R=45 m and nfe=0.3. Outliers levels are increased by steps of 10% from 0% to 50%.

**Figure 12 sensors-21-02324-f012:**
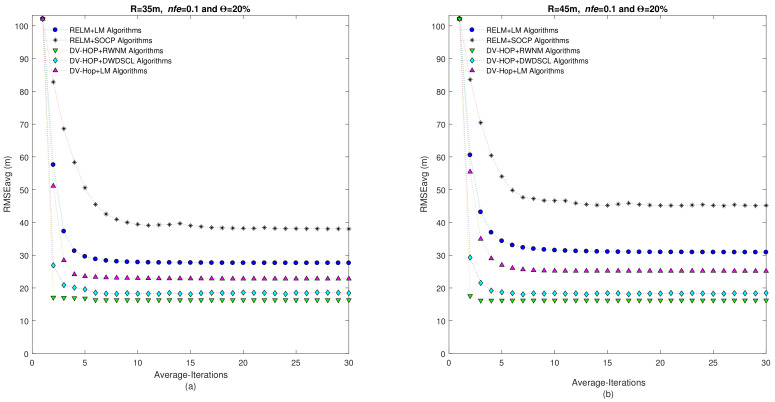
RMSEavg vs. Iterations of algorithms using Θ=20% and nfe=0.1 considering the average of 10 independent anisotropic networks. (**a**) Corresponds to R = 35 m and (**b**) R = 45 m.

**Table 1 sensors-21-02324-t001:** Simulation parameters of the experiments.

Parameters	Description\Values
Standard-deviation of the distance error (nfe)	0.1, 0.3
Outlier levels (Θ)	0%, 10%, 20%, 30%, 40%, 50%
Radio Range	35 m, 45 m
Number of Networks	10
Network Topologies	isotropic: square-shape
	anisotropic: circular-shape
Sensor Nodes Distribution	uniformly random distributed
Anchor Nodes	5
Unknown Sensors	95
Deployment Area	200 × 200 m${}^{2}$
Performance Metrics	RMSE, Iterations

**Table 2 sensors-21-02324-t002:** RMSEavg and Iter_avg at different outliers values using R = 35 m and nfe=0.1, the best performances in RMSE average and iteration average are denoted in bold face.

Algorithms	RMSEavg (m)	Iter_avg
RELM+SOCP	42.15	15.63
RELM+LM	33.98	8.7
DV-HOP+DWDSCL	**23.36**	12.81
DV-HOP+LM	30.70	9.01
DV-HOP+RWNM	24.04	**4.25**

**Table 3 sensors-21-02324-t003:** RMSEavg and Iter_avg at different outliers values using R = 35 m and nfe=0.3, the best performances in RMSE average and iteration average are denoted in bold face.

Algorithms	RMSEavg (m)	Iter_avg
RELM+SOCP	42.25	16.01
RELM+LM	36.33	9.1
DV-HOP+DWDSCL	**24.04**	9.7
DV-HOP+LM	33.17	10.16
DV-HOP+RWNM	24.84	**4.06**

**Table 4 sensors-21-02324-t004:** RMSEavg and Iter_avg at different outliers values using R = 45 m and nfe=0.1, the best performances in RMSE average and iteration average are denoted in bold face.

Algorithms	RMSEavg (m)	Iter_avg
RELM+SOCP	42.57	14.66
RELM+LM	35.05	10.53
DV-HOP+DWDSCL	14.80	10.98
DV-HOP+LM	27.31	9.75
DV-HOP+RWNM	**14.78**	**5.05**

**Table 5 sensors-21-02324-t005:** RMSEavg and Iter_avg at different outliers values using R = 45 m and nfe=0.3, the best performances in RMSE average and iteration average are denoted in bold face.

Algorithms	RMSEavg (m)	Iter_avg
RELM+SOCP	42.86	14.33
RELM+LM	39.03	10.01
DV-HOP+DWDSCL	16.13	12.93
DV-HOP+LM	32.12	10.33
DV-HOP+RWNM	**15.22**	**5.45**

**Table 6 sensors-21-02324-t006:** RMSEavg and Iter_avg at different levels of outliers Θ from 0% to 50%, R = 35 m and nfe=0.1 under anisotropic networks, the best performances in RMSE average and iteration average are denoted in bold face.

Algorithms	RMSEavg (m)	Iter_avg
RELM+SOCP	37.89	15.35
RELM+LM	28.33	10.06
DV-HOP+DWDSCL	19.26	11
DV-HOP+LM	23.65	8.08
DV-HOP+RWNM	**16.49**	**4.15**

**Table 7 sensors-21-02324-t007:** RMSEavg and Iter_avg at different levels of outliers Θ from 0% to 50%, R=35 m and nfe=0.3 under anisotropic networks, the best performances in RMSE average and iteration average are denoted in bold face.

Algorithms	RMSEavg (m)	Iter_avg
RELM+SOCP	38.23	15.53
RELM+LM	30.34	10.1
DV-HOP+DWDSCL	19.62	11.83
DV-HOP+LM	25.73	8.08
DV-HOP+RWNM	**16.76**	**4.48**

**Table 8 sensors-21-02324-t008:** RMSEavg and Iter_avg at different levels of outliers Θ from 0% to 50%, R = 45 m and nfe=0.1 under anisotropic networks, the best performances in RMSE average and iteration average are denoted in bold face.

Algorithms	RMSEavg (m)	Iter_avg
RELM+SOCP	45.41	13.75
RELM+LM	31.87	11.03
DV-HOP+DWDSCL	19.37	10.56
DV-HOP+LM	26.85	9.40
DV-HOP+RWNM	**16.24**	**5.33**

**Table 9 sensors-21-02324-t009:** RMSEavg and Iter_avg at different levels of outliers Θ from 0% to 50%, R = 45 m and nfe=0.3 under anisotropic networks, the best performances in RMSE average and iteration average are denoted in bold face.

Algorithms	RMSEavg (m)	Iter_avg
RELM+SOCP	46.22	14.01
RELM+LM	34.68	11.30
DV-HOP+DWDSCL	19.90	13.53
DV-HOP+LM	29.65	9.96
DV-HOP+RWNM	**16.75**	**5.2**

## Data Availability

The data presented in this work are available on request from the corresponding author.

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
