# Peer review of "A Weighted and Distributed Algorithm for Range-Based Multi-Hop Localization Using a Newton Method"

_sensors, 2021, doi:10.3390/s21072324_

Round 1
Reviewer 1 Report
The work addressed the problem of range-based multi-hop localization in both isotropic and anisotropic networks. The authors propose a Newton-based method using a hop-weighted function and
a scaling factor which results in good performance, from both localization and convergence point of views. The manuscript is interesting, but has room for significant improvement. I list my specific comments below.
1) Related work needs to be improved. There are a lot of existing works which deal with range-based localization that deserve to be included in the work and properly summarized (at least
some of the most recent ones).
2) The authors are assuming that all data have the same dispersion. This should be explicit after (2), since (3) would not hold if this was not the case.
3) I don't see the point of introducing sets S, U, P, A, and Q in Section 3. It seems to me that two (three if the authors really want to define U) are sufficient. Just introduce the
coordinates in sets S and A directly.
4) Just before (7), by "neighboring sensors", are the authors talking about direct neighbors?
5) What exactly does it mean "all ITS anchors"? Are you talking about all anchors in the network or some local portion of anchors? What if a sensors does not have an anchor as its direct
neighbor?
6) Is c a vector or a scalar? After (14), the authors are talking about c_i, so I assume that c is a vector. If so, it should be in bold.
7) How do you calculate (16) and (17), when it depends on unknown positions? I didn't get this part, so I would advise revision here in order to make it more clear to the reader.
8) Why are distance measurements generated according to (29) and (30) instead of (9)? Why is the true distance multiplied with e_ij?
9) How is "nfe/\Ttheta \in [0, 1]" given in percentage? Perhaps it can represent the probability or a ration, but it is certainly not given in %.
10) Don't you need to normalize (32) with the number of simulation runs that was performed, or the authors performed a single simulation run?
11) Explanation of state-of-the-art methods is needed in the Introduction section at least, in order for the reader to know what has been done so far, and to understand what are the
differences (contribution) of the present work.
12) I would say that DV-HOP+RWNM algorithm has the best overall performance in isotropic networks as well, since it practically matches the localization performance f DV-HOP+DWDSCL method,
but needs far less iterations to converge.
13) I would like to see a figure illustrating RMSE vs. number of iterations of each algorithm in order to see how these improve their performance through iterations.
14) Analysis of computational complexity and running time is missing.
15) Is there any evidence that the proposed algorithm will converge and to what? Some theoretical analysis would be nice to see in the manuscript.
16) The contribution of the work is not clear. I invite the authors to clearly state the contribution of their work in the Introduction section.
Author Response
Dear reviewer,
We appreciate your willingness and time to revise the manuscript. We have made a great effort to correct everything as you suggested. We hope you find the paper acceptable for publication.
Sincerely,
The authors.

Reviewer 2 Report
The paper presents a range-based multi-hop localization algorithm for wireless sensor networks (WSNs), which exploits the Newton scheme. Computer simulation is performed to evaluate the performance of the proposed algorithm. However, there are some flaws in the current manuscript.
It is highly suggested to discuss the pros and cons of the proposed localization algorithm. In particular, the competitive advantages over the existing localization methods should be clearly addressed.
The symbols and notations should be summarized in a table for readers to follow up the equaions more easily. Some missed definitions of symbols should also be clearly defined.
In Algorithm 1, the input and output are not explicitly given. They are necessary for clarification.
In the performance study, the five anchor nodes are equally distributed around the circle shape and uniformly distributed over the square area. If it is a strict assumption in the proposed scheme, this should be clearly addressed as an assumption in the algorithm design. Anyway, what happens if the five anchors are not equally/uniformly distributed?
More on the simulator should be clearly described and explained in detail. All the simulation parameters should be summarized in a table for readers, with some discussion.
Author Response
We thank for the valuable time spent reading and making constructive comments on this MS. We have followed all the comments, and the corresponding responses are shown after each comment. We hope the reviewer will find the revised version of our manuscript appropriate for publication in Sensors

Reviewer 3 Report
1) I am not completely satisfied with the review of the existing literature provided in the Introduction. Specifically, the authors highlight that a localization algorithm should deal with realistic multi-hop WSN scenarios, in which hardware imperfections as well as changing environments could be present. However, the cited literature does not address these specific problems, but only deals with anisotropic networks. In this respect, I would suggest adding some proper references about localization in challenging scenarios such as:
- G. Ricci et al, "A Pseudo Maximum likelihood approach to position estimation in dynamic multipath environments", Signal Processing, Vol. 181, April 2021;
- D. Humphrey et al, "WASP: A System and Algorithms for Accurate Radio Localization Using Low-Cost Hardware," in IEEE Transactions on Systems, Man, and Cybernetics, Part C (Applications and Reviews), vol. 41, no. 2, pp. 211-222, March 2011
but authors are encouraged to add also other references they are aware of.
2) After eq. (2), it should be "the joint probability density function" instead of "the joint density function".
3) In eqs. (1), (3), (4), and (6), please add the optimization variable underneath the corresponding "arg min" or "arg max" operator;
4) Why is eq. (11) defined as the distance error with one-hop neighboring sensors? Isn't each anchor in the set B_i (eq. (8)) reachable with n-hops by sensor s_i?
5) After eq. (9), please specify whether e_ij is deterministic or random, and provide the proper details accordingly.
6) Eq. (13) should be the set of the weights corresponding to the neighbors of sensor s_i, not the "weighted-hop set of neighboring sensors of s_i" as it is currently stated in the paper.
7) Since the sensors have unknown positions, how can one compute the elements in the set in eq. (16) required, in turn, to compute the median in eq. (15)?
8) "Assuming that rough initial estimates are provided for all unknown sensor nodes...", this is actually a strong assumption in the considered problem, where sensors are randomly deployed without any possibility to have a first coarse information of their positions. At least, one should mention how such rough estimates are obtained, given that sensors are not equipped with any position-aware device, and what is their accuracy.
9) Please fix Fig. 2, some of the labels are out of proportion compared to the others.
10) As a general final comment, please carefully proofread the whole paper since it contains a lot of grammar mistakes.
Author Response
We appreciate your willingness and time to review the manuscript. We have made a great effort to correct everything as you suggested. We hope you find the revised version of our manuscript appropriate for publication in Sensors

Round 2
Reviewer 1 Report
The authors did a good job in addressing the Reviewer's comments. I advise them to add an averaging term in (41), where they will divide the RMSE with Mc, being Mc the number of Monte Carlo runs (simulations) that they are performing, i.e., $\text{RMSE} = \sqrt{ \frac{1}{M_C N} \displaystyle\sum_{j=1}^{M_C} \displaystyle\sum_{i=1}^{N} \| \boldsymbol{p}_{ij} - \boldsymbol{u}_{ij} \|^2 }$.
Author Response
Dear reviewer,
Thank you for your comments and observation. We have made changes according to your recommendation. We have modified and improved the RMSE metric (See equation (41)), used for the evaluation of the accuracy performance of the algorithm, as you suggested. Also, we have changed Y_axis and X_axis in the corresponding Figures.

Reviewer 2 Report
The review comments have been responded appropriately and the paper has been revised accordingly. Only one more comment has been neither responded nor revised. That is, the simulation (experiment) parameters has not been summarized in a table. This is very helpful for readers to read the paper more easily.
Author Response
Thank you. We appreciate your comments and observations. According to your advice, we have included a Table (See Table 1) summarizing the experimental parameters used in our simulations.

Reviewer 3 Report
Authors correctly addressed all my comments.
Author Response
Thank you. We appreciate your comments and advice that allow us to improve the manuscript.